# Out-of-Distribution Detection with Overlap Index

## Abstract

Out-of-distribution (OOD) detection is crucial for the deployment of machine learning models in the open world. While existing OOD detectors are effective in identifying OOD samples that deviate significantly from in-distribution (ID) data, they often come with trade-offs. For instance, deep OOD detectors usually suffer from high computational costs, require tuning hyperparameters, and have limited interpretability, whereas traditional OOD detectors may have a low accuracy on large high-dimensional datasets. To address these limitations, we propose a novel effective OOD detection approach that employs an overlap index (OI)-based confidence score function to evaluate the likelihood of a given input belonging to the same distribution as the available ID samples. The proposed OI-based confidence score function is non-parametric, lightweight, and easy to interpret, hence providing strong flexibility and generality. Extensive empirical evaluations indicate that our OI-based OOD detector is competitive with state-of-the-art OOD detectors in terms of detection accuracy on a wide range of datasets while requiring less computation and memory costs. Lastly, we show that the proposed OI-based confidence score function inherits nice properties from OI (e.g., insensitivity to small distributional variations and robustness against Huber $\epsilon$-contamination) and is a versatile tool for estimating OI and model accuracy in specific contexts.

## 1 Introduction

Machine learning models often struggle with out-of-distribution (OOD) samples that originate from distributions not seen during training. The OOD detection (Macêdo et al., 2021; Zhao et al., 2023) is a critical task to address the above issue and has been utilized in many real-world applications (Zhao et al., 2023; Wang et al., 2022a). Given in-distribution (ID) samples originating from the training distribution, the OOD detector employs a confidence scorer to quantify the distance between a given input and the available ID samples to filter out OOD inputs, thereby enhancing the machine learning model's reliability in unseen environments.

Traditional OOD detectors like one-class SVM (Schölkopf et al., 2001) are computationally and memory-efficient but are relatively less effective on large high-dimensional datasets. While deep learning approaches such as Deep SVDD (Ruff et al., 2018) are capable of handling large high-dimensional datasets, they come at the cost of substantial computational and memory overhead due to their large number of parameters. Gaussian-based OOD detectors, such as (Morteza & Li, 2022), require calculating the inverse of the covariance matrix which can be numerically unstable. To address these limitations, this paper proposes a novel OOD detector that is non-parametric (i.e., no assumption about the underlying data distribution)[1], lightweight, and effective on large high-dimensional datasets.

The proposed OOD detector employs an overlap index (OI)-based confidence score function to evaluate the likelihood of a given input belonging to the same distribution as the available ID samples. OI is a widely utilized metric that quantifies the area of intersection between two probability density functions. It is closely related to the total variation distance (TVD), where TVD is one minus OI (Pastore & Calcagnì, 2019). Both OI and TVD find applications in various domains (Zhang et al., 2021b; Kapralov et al., 2016). Unlike Kullback–Leibler (KL) and Jensen–Shannon (JS) divergences, OI does not require the supports of the distributions to be identical and is insensitive to small distributional variations (Larry Wasserman, a).

---

[1] Note that non-parametric methods could still have tunable parameters.

While the Wasserstein distance requires distance inference (Yatracos, 2022) and lacks robustness to Huber $\epsilon$-contamination outliers (Huber, 1992) with unbounded metrics, OI does not suffer from these limitations. Despite the applications and advantages of OI, its utilization for OOD detection is seldom leveraged in existing literature. This paper endeavors to explore the potential applications of the OI-based confidence score function in OOD detection, showcasing its effectiveness.

Our OI-based confidence score function is converted from a novel upper bound of OI derived in this paper. The derived bound consists of two computationally efficient terms: the distance between the means of the clusters and the TVD between two distributions over subsets that meet specific conditions. The derived upper bound is quick to compute, non-parametric, insensitive to small distributional variations, and robust against Huber $\epsilon$-contamination outliers. If one cluster contains only a given input and the other cluster contains a few clean ID samples, then this bound could serve as a confidence score function for evaluating the likelihood that the given input belongs to the same distribution as the available ID samples.

The proposed OI-based OOD detector addresses the aforementioned limitations of the existing OOD detectors. Compared to traditional OOD detectors, our method is empirically effective even on large high-dimensional datasets. Compared to deep OOD detectors, our OI-based detector avoids the use of neural networks, thereby being more economical in terms of computational and memory requirements. Despite this, it achieves accuracy levels comparable to those of state-of-the-art deep OOD detectors. This makes our approach particularly beneficial in scenarios where the available ID dataset is too small to effectively train a deep learning-based OOD detector. Another group of OOD detectors is Gaussian-based OOD detectors, such as GEM (Morteza & Li, 2022), MSP (Hendrycks & Gimpel, 2017), Mahalanobis distance (Lee et al., 2018), and energy score (Liu et al., 2020). These methods assume Gaussian distributions and apply distance-based metrics on deep features. Compared to them, our method neither makes distributional assumptions about the underlying feature space nor requires the computationally challenging and numerically unstable task of inverting covariance matrices. Zhang et al. (2023) uses an KL-based confidence scorer for OOD detection, which also assumes Gaussian distribution for the extracted features.

We show that the variant of our OI-based confidence scorer could be employed for estimating both OI and model accuracy in specific contexts. Sriperumbudur et al. (2012) estimates OI using integral probability metrics (IPM)[2], which results in a computational complexity on the order of $\mathcal{O}^*(n^\omega)$, where $\omega$ is the matrix multiplication exponent (Cohen et al., 2021). Additionally, selecting appropriate functions for IPM-based methods often presents a challenge (Larry Wasserman, b). In contrast, our method approximates OI using a class of straightforward, easily identifiable, and computationally efficient conditional functions[3]. Schmid & Schmidt (2006); Pastore & Calcagnì (2019) estimate OI using estimated probability density functions through kernel methods[4]. These methods are burdened by the curse of dimensionality (Zambom & Ronaldo, 2013), whereas our approach remains efficient in high-dimensional spaces.

Overall, the contributions of this paper include: 1) proposing a novel OOD detector that leverages an OI-based confidence score function; 2) evaluating the proposed OOD detector with various state-of-the-art methods and datasets to demonstrate its effectiveness and efficiency; 3) analyzing the mathematical properties of the OI-based confidence score function; and 4) exploring into the potential applications of the OI-based confidence score function for estimating OI and model accuracy in specific contexts.

The remaining part is organized as follows: We first conduct a literature review on OI and OOD detection in Sec. 2. Then Sec. 3 derives the novel upper bound for OI and proposes our novel OI-based OOD detector. Sec. 4 empirically evaluates the proposed novel OOD detector. Sec. 5 provides mathematical analysis and the potential applications of our approach for estimating OI and model accuracy. Sec. 6.2 discusses the limitations and broader impact of our approach. Sec. 7 concludes the paper.

---

[2] $d_{\mathcal{F}}(P,Q) = \sup_{f \in \mathcal{F}} |\mathbb{E}X \sim P[f(X)] - \mathbb{E}Y \sim Q[f(Y)]|$, where $\mathcal{F}$ is a class of functions.

[3] The condition function $\mathbb{1}\{\cdot\}$ outputs 1 when the input satisfies the given condition and 0 otherwise.

[4] $\hat{f}(x) = \frac{1}{n}\sum_{i=1}^{n} \mathcal{K}\left(\frac{x-x_i}{\beta}\right)$, where $\mathcal{K}$ is the kernel function and $\beta$ is the bandwidth.

## 2 Background and Related Works

### 2.1 OOD Detection

A comprehensive discussion of classic OOD detectors, such as one-class SVM (Schölkopf et al., 2001), decision-tree (Comité et al., 1999), and one-class nearest neighbor (Tax, 2002), is given by Khan & Madden (2014). The deep OOD detectors can be divided into image-level OOD detectors and feature-level OOD detectors. The image-level OOD detectors train their models using raw inputs, whereas the feature-level OOD detectors require pretrained models to process the input. For image-level OOD detectors, Deep SVDD (Ruff et al., 2018), OCGAN (Perera et al., 2019), and GradCon (Kwon et al., 2020) do not use additional information, whereas Deep SAD (Ruff et al., 2020) uses information from anomaly samples. Bergmann et al. (2020) uses a teacher network for OOD detection. CutPaste (Li et al., 2021a) considers object detection. Schneider et al. (2022) studies different autoencoders for OOD detection. For feature-level OOD detectors, You et al. (2023) achieves a higher accuracy using a transformer instead of autoencoders to reconstruct feature maps. Golan & El-Yaniv (2018) bypasses the feature reconstruction phase using self-labeled training datasets. LOE (Qiu et al., 2022), Panda (Reiss et al., 2021), and Salehi et al. (2021) use pretrained ImageNet models to increase their accuracy. LPIPS (Zhang et al., 2018) is a feature-based similarity metric for images. ECOD (Li et al., 2022) is a non-parametric OOD detector, which calculates the confidence score using empirical cumulative distribution functions. Xu et al. (2023) proposes a deep isolation forest for OOD detection that uses neural networks to map original data into random representation ensembles. Other works aiming to improve OOD detection accuracy for classification and object detection include ViM (Wang et al., 2022b), GradNorm (Huang et al., 2021), VOS (Du et al., 2022), PatchCore (Roth et al., 2022), YolOOD (Zolfi et al., 2024). Li et al. (2023) uses masked image modeling for OOD detection. Dream-OOD (Du et al., 2023) investigates the use of diffusion models for OOD detection. Specifically, it learns a text-conditioned latent space from ID data and subsequently samples outliers in the low-likelihood region of the latent space. These outliers can then be decoded into images using the diffusion model.

### 2.2 Statistical Divergences and Estimating OI

Popular statistical divergences to measure distribution similarities contain OI, TVD, KL divergence, and Wasserstein distance. KL divergence is often utilized in classification (Malinin & Gales, 2019) and reinforcement learning (Even-Dar et al., 2006), whereas the Wasserstein distance is preferred in optimal transport (Kolouri et al., 2017; Raghvendra et al., 2024; Phatak et al., 2023; Lahn et al., 2023) and image processing (Cuturi & Doucet, 2014). Note that TVD is less than or equal to the square root of KL divergence. Therefore, many results from KL divergence can transfer to TVD and thus OI, or vice versa. Estimating OI and TVD with unknown distributions using only finite samples is challenging. The kernel-based estimations (Schmid & Schmidt, 2006; Pastore & Calcagnì, 2019) are sensitive to the choice of bandwidth, have a boundary effect, and suffer from the curse of dimensionality. We found that their methods are slow in high-dimensional space. The Cohen's d measure (Inman & Bradley Jr, 1989) is efficient but assumes that the distributions $A, B$ are Gaussian with the same standard deviation $\sigma$ and approximates $OI \approx 2\Phi(-\frac{|\mu_A - \mu_B|}{2\sigma})$, where $\mu_A$ and $\mu_B$ are samples' means and $\Phi(\cdot)$ is the standard normal distribution function. If the assumption does not hold, the method may not be accurate. Sriperumbudur et al. (2012) provides an estimator for TVD, which requires solving time-consuming linear programs and is not consistently accurate, as noted by the authors. Our OI-based confidence score function shows a potential for estimating OI in a time-efficient fashion without suffering from the curse of dimensionality.

## 3 The OI-Based OOD Detector

### 3.1 Problem Formulation and Goals

Given $\mathbb{R}^n$ space and $m$ samples $\{x_i\}_{i=1}^m$ that lie in an unknown distribution, we would like to build a binary detector $\Psi : \mathbb{R}^n \to \{\pm 1\}$ such that for any new input $x$, $\Psi(x)$ outputs 1 when $x$ is from the same unknown probability distribution, and outputs -1, otherwise. The detector $\Psi$ should be effective even when $n$ is large or $m$ is small. Additionally, the computation and memory costs of $\Psi$ should be small.

### 3.2 Motivation: A Novel Upper Bound for OI

Our methodology is inspired by a newly derived upper bound for the OI between bounded distributions. We consider the $\mathbb{R}^n$ space and continuous random variables. We define $P$ and $Q$ as two probability distributions in $\mathbb{R}^n$ with $f_P$ and $f_Q$ being their probability density functions. The derivation of this novel upper bound necessitates the introduction of the following definitions.

**Definition 3.1.** The OI, $\eta$, is a function of two distributions whose samples belong to the $\mathbb{R}^n$ space and outputs the overlap index value of these two distributions, which is a real number between 0 and 1. $\eta$ is defined as (Pastore & Calcagnì, 2019):

$$\eta(P, Q) = \int_{\mathbb{R}^n} \min[f_P(x), f_Q(x)] dx. \tag{1}$$

**Definition 3.2.** Given an $A \subset \mathbb{R}^n$, $\delta_A$ is a function of two distributions whose samples belong to the $\mathbb{R}^n$ space and outputs a real number between 0 and 1 with the following definition:

$$\delta_A(P, Q) = \frac{1}{2} \int_A |f_P(x) - f_Q(x)| dx. \tag{2}$$

The standard TVD is $\delta_A$ with $A = \mathbb{R}^n$ or one minus OI (Pastore & Calcagnì, 2019). The following theorem shows a novel upper bound of OI between bounded distributions.

**Theorem 3.3.** *Without loss of generality, assume $D^+$ and $D^-$ are two probability distributions on a bounded domain $B \subset \mathbb{R}^n$ with defined norm[5] $||\cdot||$ (i.e., $\sup_{x \in B} ||x|| < +\infty$), then $\forall\ A \subset B$ with $A^c = B \setminus A$, we have*

$$\eta \leq 1 - \frac{1}{2r_{A^c}} ||\mu_{D^+} - \mu_{D^-}|| - \frac{r_{A^c} - r_A}{r_{A^c}} \delta_A \tag{3}$$

*where $r_A = \sup_{x \in A} ||x||$ and $r_{A^c} = \sup_{x \in A^c} ||x||$, $\mu_{D^+}$ and $\mu_{D^-}$ are the means of $D^+$ and $D^-$, and $\delta_A$ is TVD on set $A$ as defined in* **Definition** *3.2. Moreover, let $r_B = \sup_{x \in B} ||x||$, then*

$$\eta \leq 1 - \frac{1}{2r_B} ||\mu_{D^+} - \mu_{D^-}|| - \frac{r_B - r_A}{r_B} \delta_A. \tag{4}$$

*Since (4) holds for any $A$, a tighter bound can be written as*

$$\eta \leq 1 - \frac{1}{2r_B} ||\mu_{D^+} - \mu_{D^-}|| - \max_A \frac{r_B - r_A}{r_B} \delta_A. \tag{5}$$

The theorem is framed to accommodate probability distributions with bounded domains. In practical applications, data may originate from unbounded distributions. Nevertheless, the corresponding finite-sample datasets are inherently bounded. As such, it is possible to approximate the underlying unbounded distributions with bounded ones and then apply our theorem. We forego further exploration of cases involving unbounded distributions at this time, as the current theorem sufficiently informs the intuition behind our proposed OOD detector. To calculate $\delta_A$ and $||\mu_{D^+} - \mu_{D^-}||$ with finite samples, we have the following corollary.

**Corollary 3.4.** *Given $D^+$, $D^-$, $B$, and $||\cdot||$ used in* **Theorem** *3.3, let $A(g) = \{x \mid g(x) = 1, x \in B\}$ with any condition function $g : B \to \{0, 1\}$. An upper bound for $\eta(D^+, D^-)$ can be obtained:*

$$\eta \leq \overline{\eta} = 1 - \frac{1}{2r_B} ||\mu_{D^+} - \mu_{D^-}|| - \max_g \frac{r_B - r_{A(g)}}{2r_B} \left| \mathbb{E}_{D^+}[g] - \mathbb{E}_{D^-}[g] \right|. \tag{6}$$

$|\mathbb{E}_{D^+}[g] - \mathbb{E}_{D^-}[g]|$ is the absolute value. If we use only a single input to calculate $\mu_{D^+}$ and $\mathbb{E}_{D^+}[g]$ and use a few samples to calculate $\mu_{D^-}$ and $\mathbb{E}_{D^-}[g]$, then $\overline{\eta}$ can be considered as a confidence score function for evaluating the likelihood that the given input in $D^+$ belongs to the same distribution as samples in $D^-$. Before providing further visualization and ablation study to demonstrate the efficacy of using such a confidence score function, we present the implementation algorithm to calculate equation 6.

---

[5]This paper considers the $L_2$ norm. However, the analysis can be carried out using other norms.

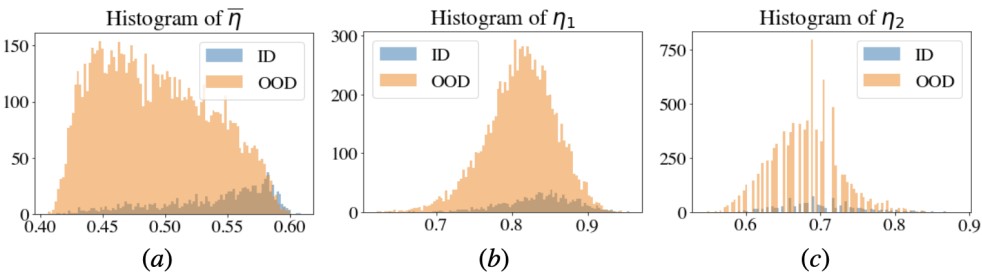

Figure 1: Histograms of confidence scores using $\overline{\eta}$, $\eta_1$, and $\eta_2$ with plane as the ID class and the other nine classes as the OOD class in CIFAR-10. $D^+$ consists of all samples from the "plane" class, while $D^-$ includes all samples from the remaining nine classes.

### 3.3 Empirical Calculation of the Novel Upper Bound

The middle term in the RHS of equation 6 is the distance of clusterings' means, and the last term is a function of IPM with a class of condition functions. To approximate the last term, we draw samples from $D^+$ and $D^-$ and then average their $g$ values to estimate $\mathbb{E}_{D^+}[g]$ and $\mathbb{E}_{D^-}[g]$. Alg. 1 shows the computation of $\overline{\eta}$ with given $k$ condition functions $\{g_j\}_{j=1}^k$ and finite sample, $\{x_i^+\}_{i=1}^d \sim D^+$ and $\{x_i^-\}_{i=1}^m \sim D^-$.

---

**Algorithm 1** ComputeBound($\{x_i^+\}_{i=1}^d$, $\{x_i^-\}_{i=1}^m$, $\{g_j\}_{j=1}^k$)

---

$B \leftarrow \{x_1^+, ..., x_d^+, x_1^-, ..., x_m^-\}$
$r_B \leftarrow \max_{x \in B} \|x\|$
$\Delta_\mu \leftarrow \left\| \frac{1}{d} \sum_{i=1}^d x_i^+ - \frac{1}{m} \sum_{i=1}^m x_i^- \right\|$
**FOR** $j = 1 \rightarrow k$
  $A(g_j) \leftarrow \{x \mid g_j(x) = 1, x \in B\}$
  $r_A \leftarrow \max_{x \in A} \|x\|$
  $s_j \leftarrow (r_B - r_A) \left| \frac{1}{d} \sum_{i=1}^d g_j(x_i^+) - \frac{1}{m} \sum_{i=1}^m g_j(x_i^-) \right|$
**Return:** $1 - \frac{1}{2r_B} \Delta_\mu - \frac{1}{2r_B} \max_j s_j$

---

**Choice of $g$:** The choice of condition functions is not unique. The literature faces similar issues when choosing functions for IPMs (Larry Wasserman, b; Sriperumbudur et al., 2012). The overall guideline is to choose functions that satisfy convenient regularity conditions. The chosen functions should also be computation-efficient and memory-efficient. Considering all the aspects, $g_j(x) = \mathbb{1}\{r_{j-1} \leq \|x\| \leq r_j\}$ is selected with $r_j = jr_B/k$, which are nicely regularized, computation-efficient, and empirically effective. However, other condition functions are worth exploring.

**The OI-Based Confidence Score Function**: The confidence score function using Alg. 1 is defined as $f(x) = ComputeBound(\{x\}, \{x_i\}_{i=1}^m, \{g_j\}_{j=1}^k)$ (i.e., $\{x^+\}^d = \{x\}$ with $d = 1$), which measures the maximum similarity between $x$ and $\{x_i\}_{i=1}^m$. If $f(x) \geq T_0$, then $x$ is considered as an ID sample. Otherwise, $x$ is considered as OOD. $T_0$ is pre-defined by users.

### 3.4 Computation and Space Complexities

We can pre-compute and store $\frac{1}{m} \sum_{i=1}^m x_i$ and $\frac{1}{m} \sum_{i=1}^m g_j(x_i)$ in Alg. 1. Therefore, the space complexity is $\mathcal{O}(k+1)$. The computation complexity is $\mathcal{O}(k+1)$ for each online input $x$ since it needs to calculate $\|x\|$ once and $s_j$ for $k$ times. $k$ can be restricted to a reasonable number (e.g., $\leq 100$) so that even devices without strong computation power (e.g., Arduino or Raspberry Pi) can run our OOD detector efficiently. This is an advantage compared to time-consuming deep approaches.

**Visualization**: Although our confidence score seems simple, we empirically find it effective. Both $\frac{1}{2r_B}\|\mu_{D^+} - \mu_{D^-}\|$ and $\max_g \frac{r_B - r_{A(g)}}{2r_B}|\mathbb{E}_{D^+}[g] - \mathbb{E}_{D^-}[g]|$ are important for our OOD detector. To validate their efficacy, we created two another OOD detectors with $\eta_1 = 1 - \frac{1}{2r_B}\|\mu_{D^+} - \mu_{D^-}\|$ and

Table 1: Information on utilized UCI datasets.

| Dataset | $n$ | ID Class | Size | OOD Class | Size |
|---|---|---|---|---|---|
| Iris | 4 | Setosa | 50 | Vers.+Virg. | 100 |
| | | Versicolour | 50 | Seto.+Virg. | 100 |
| | | Virginica | 50 | Seto.+Vers. | 100 |
| Breast Cancer | 9 | Malignant | 241 | Beni. | 458 |
| | | Benign | 458 | Mali. | 241 |
| Ecoli | 7 | Peripalsm | 52 | All Others | 284 |
| Ball-bearing | 32 | Ball-bearing | 913 | None | 0 |

$\eta_2 = 1 - \max_g \frac{r_B - r_{A(g)}}{2r_B} |\mathbb{E}_{D^+}[g] - \mathbb{E}_{D^-}[g]|$ as the confidence score functions. Fig. 1 shows the histograms of $\eta$, $\eta_1$, and $\eta_2$ using CIFAR-10 (Krizhevsky et al., 2009) plane images as the ID samples and the other nine class images as the OOD samples. Fig. 1(b,c) show that it is less likely to distinguish between ID and OOD samples by independently observing either $\frac{1}{2r_B}||\mu_{D^+} - \mu_{D^-}||$ or $\max_g \frac{r_B - r_{A(g)}}{2r_B} |\mathbb{E}_{D^+}[g] - \mathbb{E}_{D^-}[g]|$. However, the summation of $\frac{1}{2r_B}||\mu_{D^+} - \mu_{D^-}||$ and $\max_g \frac{r_B - r_{A(g)}}{2r_B} |\mathbb{E}_{D^+}[g] - \mathbb{E}_{D^-}[g]|$ amplifies the difference between ID and OOD samples, as shown in Fig. 1(a). Our OOD detector successfully utilizes the dependence between $\frac{1}{2r_B}||\mu_{D^+} - \mu_{D^-}||$ and $\max_g \frac{r_B - r_{A(g)}}{2r_B} |\mathbb{E}_{D^+}[g] - \mathbb{E}_{D^-}[g]|$ to identify OOD samples.

**Flexibility**: Our OOD detector does not require any distributional assumptions nor the need to calculate the inverse of the covariance matrix which can be numerically unstable. Besides, Alg. 1 could be applied in any space, such as the feature space generated by a pretrained model. This flexibility broadens the potential application of our approach. Additionally, Alg. 1 supports batch-wise computation since $\frac{1}{m}\sum_{i=1}^{m} x_i$ and $\frac{1}{m}\sum_{i=1}^{m} g_j(x_i)$ can be pre-computed and stored.

## 4 Experimental Results

### 4.1 Setup

**Datasets**: We employ UCI datasets (Dua & Graff, 2017) given in Table 1 for benchmarking against traditional OOD detectors. For comparison with state-of-the-art deep OOD detectors, we primarily use the CIFAR-10 dataset (Krizhevsky et al., 2009) in both input and feature spaces. Additionally, the CIFAR-100 dataset (Krizhevsky et al., 2009) is utilized to serve as the ID dataset for comparisons with Gaussian-based OOD detectors. OOD datasets include Textures (Cimpoi et al., 2014), SVHN (Netzer et al., 2011), LSUN-Crop (Yu et al., 2015), LSUN-Resize (Yu et al., 2015), and iSUN (Xu et al., 2015). We also extend the application of our OOD detector to backdoor detection, leveraging datasets such as MNIST (LeCun et al., 2010), GTSRB (Stallkamp et al., 2011), YouTube Face (Wolf et al., 2011), and sub-ImageNet (Deng et al., 2009). Details on these datasets are provided in Appendix F.

**Metrics**: The primary metric employed for assessing the efficacy of our OOD detectors is the area under the receiver operating characteristic curve (AUROC). A higher AUROC value indicates better accuracy in differentiating between ID and OOD samples. Supplementary to AUROC, we also utilize TPR95—representing the detection accuracy for OOD samples when the detection accuracy for ID samples is fixed at 95%—and the area under the precision-recall curve (AUPR) as additional performance metrics.

**Hyperparameters**: We set $k = 100$ and $g_j(x)$ with the form $\mathbb{1}\{r_{j-1} \le ||x|| \le r_j\}$ in Alg. 1. We recommend choosing $k$ between 50 and 200. Values of $k \ge 200$ do not significantly improve performance, while $k < 50$ results in underperformance. The selected condition functions $g_j$ are both computationally efficient and empirically effective. The number $m$ of available ID samples varies depending on the dataset in use. We perform ablation studies to evaluate the impact of these hyperparameters on our method's performance.

**Hyperparameters of compared methods**: We obtained the code for each comparison method from the authors' respective websites and followed the provided instructions to run their approach to replicate the results reported in their original papers. Some baselines require multiple runs, and we executed them 10 times.

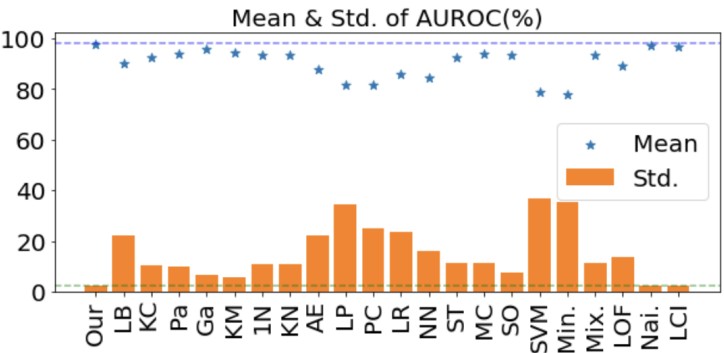

Figure 2: AUROC on UCI datasets. Horizontal lines: the mean and standard deviation of our approach.

Table 2: AUROC (%) on CIFAR-10 with each class being ID. **Boldface** shows the highest AUROC.

| Class | Raw Image Space | | | | | | Extra Information | | | | |
|---|---|---|---|---|---|---|---|---|---|---|---|
| | Ours | OCGan | Deep SVDD | AnoGAN | DCAE | GradCon | Ours | LPIPS | Bergman | PANDA | ADTR |
| Plane | **76.7** | 75.7 | 61.7±4.1 | 67.1±2.5 | 59.1±5.1 | 76.0 | 96.1 | 79.3 | 78.9 | 93.9 | **96.2** |
| Car | **67.0** | 53.1 | 65.9±2.1 | 54.7±3.4 | 57.4±2.9 | 59.8 | 97.5 | 94.6 | 84.9 | 97.1 | **98** |
| Bird | 53.4 | 64.0 | 50.8±0.8 | 52.9±3.0 | 48.9±2.4 | **64.8** | 94 | 63.1 | 73.4 | 85.4 | **94.5** |
| Cat | 58.5 | **62.0** | 59.1±1.4 | 54.5±1.9 | 58.4±1.2 | 58.6 | **94.3** | 73.7 | 74.8 | 85.4 | 91.7 |
| Deer | 73.0 | 72.3 | 60.9±1.1 | 65.1±3.2 | 54.0±1.3 | **73.3** | **95.7** | 72 | 85.1 | 93.6 | 95.1 |
| Dog | **66.2** | 62.0 | 65.7±2.5 | 60.3±2.6 | 62.2±1.8 | 60.3 | **96.4** | 75.2 | 79.3 | 91.2 | 95.6 |
| Frog | **73.3** | 72.3 | 67.7±2.6 | 58.5±1.4 | 51.2±5.2 | 68.4 | 97.9 | 83.6 | 89.2 | 94.3 | **98** |
| Horse | 58.8 | 57.5 | **67.3±0.9** | 62.5±0.8 | 58.6±2.9 | 56.7 | 94.9 | 81.8 | 83 | 93.5 | **97.1** |
| Ship | 78.0 | **82.0** | 75.9±1.2 | 75.8±4.1 | 76.8±1.4 | 78.4 | 97.9 | 81.2 | 86.2 | 95.1 | **98** |
| Truck | **74.7** | 55.4 | 73.1±1.2 | 66.5±2.8 | 67.3±3.0 | 67.8 | **97.1** | 86.3 | 84.8 | 95.2 | 96.9 |
| Ave. | **68.0** | 65.6 | 64.8 | 61.8 | 59.4 | 66.4 | **96.2** | 79.1 | 82.0 | 92.5 | 96.1 |

For instances where we applied the compared methods to new datasets, we conducted careful hyperparameter tuning to ensure their best performance.

## 4.2 Comparison Results

**UCI Dataset**: We first evaluated our OI-based detector on small UCI datasets given in Table 1. Fig. 2(a) shows that our detector outperforms all baseline methods by showing the highest average AUROC (i.e., 97.67±2.55%). Table 10 in the appendix provides the numerical numbers.

**CIFAR-10 Dataset**: We then use CIFAR-10 images with one class being ID and the remaining classes being OOD. The compared deep OOD detectors on raw images without using additional information from anomaly samples include DCAE (Makhzani & Frey, 2015), AnoGAN (Schlegl et al., 2017), Deep SVDD (Ruff et al., 2018), OCGAN (Perera et al., 2019), and GradCon (Kwon et al., 2020). Comparisons with more recent approaches are provided later. All the methods are given $m = 5000$ available ID samples. The result is in Table 2 "Raw Image Space". Our approach shows the highest AUROC. Although the baseline methods could reach similar accuracy, they are slower than ours on the same CPU machine. For example, the Deep SVDD requires 446.8 ms for each detection (ours is 3 ms). Fig. 3(a) shows the AUROC of our approach with $m$ less than 5000. With only $m = 100$ ID samples, the AUROC of our approach is 67.2%. In contrast, the AUROC of Deep SVDD drops to 61.1% with $m = 100$. As for the memory cost, Deep SVDD uses LeNet (LeCun et al., 1998) with three convolutional layers. The first layer already has 2400 parameters, whereas our approach has $\mathcal{O}(k + 1)$ space complexity with $k = 100$.

**Ablation Study**: We compare our approach with classic OOD detectors that are fit by samples' norms, including LOF (Breunig et al., 2000), OCSVM (Schölkopf et al., 2001), Isolation Forest (Liu et al., 2008), and Elliptic Envelope (Rousseeuw & Driessen, 1999). Table 3 shows that our approach is more than just calculating the statistic of the sample's norms. When using $\eta_1$ and $\eta_2$ mentioned in Sec. 3 as the confidence score functions, the AUROC of our approach becomes 58.4% for $\eta_1$ and 50.1% for $\eta_2$. These two numbers

Table 3: AUROC of our approach using $\eta_1$ and $\eta_2$ as the confidence score functions and AUROC of other methods with samples' norms as inputs.

| Method | $\eta_1$ | $\eta_2$ | LOF | OCSVM | Ellip. Env. | Iso. Fo |
|--------|----------|----------|-----|-------|-------------|---------|
| AUROC | 58.4 | 50.1 | 60.5 | 55.2 | 55.2 | 56.7 |

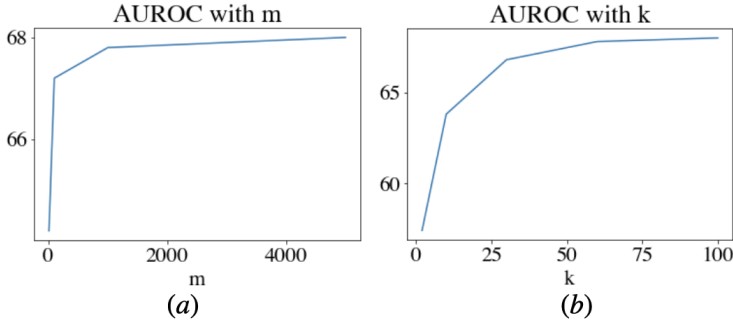

Figure 3: AUROC of our approach with (a) different numbers $m$ of available ID samples and (b) different numbers $k$ of condition functions.

are consistent with Fig. 1 that either term cannot individually detect OOD samples. Fig. 3(b) shows the AUROC of our approach with different $k$. Our approach remains high AUROC when $k \geq 50$.

**Runtime Efficiency**: ECOD (Li et al., 2022) is also non-parametric and does not require training parameters. The time complexity of ECOD is $\mathcal{O}(n)$ that increases with the data dimension $n$. We compare the execution time of our approach with ECOD by varying the data dimension $n$. We also report the computation time of Deep Isolation Forest (Xu et al., 2023). Table 4 shows the execution time. The computation complexity of our method and Deep Isolation Forest do not increase with the data dimension, whereas the computation complexity of ECOD increases linearly with the data dimension. The time complexity of our approach increases linearly with the number $k$ of utilized condition functions. We empirically observed that with $k = 1000$, the execution time becomes 36 ms.

**Improvement with Extra Information**: Deep OOD detectors such as LPIPS (Zhang et al., 2018), Bergmann's method (Bergmann et al., 2020), Panda (Reiss et al., 2021), and ADTR (You et al., 2023) require pretrained models or additional information from anomaly samples to identify OOD samples. If our approach is also allowed to use pretrained models and information from anomaly samples, the accuracy will improve. Specifically, we use a pretrained ImageNet model from (Reiss et al., 2021) to extract CIFAR-10 features. We also assume that a contaminated dataset is available. Similar assumption is also used in (Choi et al., 2024). To build the contaminated dataset, we first merge all ID and OOD samples and then randomly select 100 samples. During the testing, each input will be subtracted from the mean vector of this contaminated dataset. The results are shown in Table 2 "Extra Information". The average AUROC of our approach increases to 96.2%. Fig. 4(a) shows the improvement of our approach. According to Fig. 4(b), $\max_g \frac{r_B - r_{A(g)}}{2r_B} |\mathbb{E}_{D^+}[g] - \mathbb{E}_{D^-}[g]|$ helps distinguish between ID samples and OOD samples after using the pretrained model and the contaminated dataset. Although the compared methods may show similar performance, their training time (e.g., PANDA) is much higher than ours. We have further compared our

Table 4: Execution time (ms) per sample with various data dimension.

| $n$ | 10 | 100 | 500 | 1000 | 2000 |
|-----|-----|-----|-----|------|------|
| Ours | 3 | 3 | 3 | 3 | 3 |
| ECOD | 8 | 79 | 453 | 1021 | 2245 |
| Deep Isolation Forest | 150 | 150 | 150 | 150 | 150 |

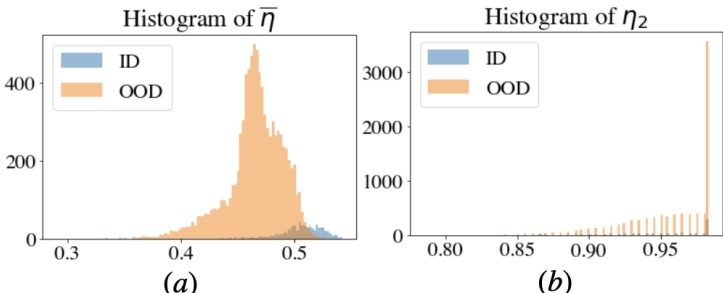

Figure 4: Histograms of confidence scores using $\overline{\eta}$ and $\eta_2$ with plane as the ID class and the other nine classes as the OOD class in CIFAR-10.

Table 5: Results for CIFAR-10 and CIFAR-100 being in-distribution datsets. **Boldface** shows the best performance, whereas underline shows the second best.

| Out-of-Distribution Datasets | Method | ID Dataset: CIFAR-10 | | | ID Dataset: CIFAR-100 | | |
|---|---|---|---|---|---|---|---|
| | | TPR95 (%) | AUROC (%) | AUPR (%) | TPR95 (%) | AUROC (%) | AUPR (%) |
| Texture | Ours | 64.20 | 92.80 | 92.33 | 42.50 | 85.79 | 85.27 |
| | MSP | 40.75±3.95 | 88.31±0.93 | 97.08±1.32 | 16.71±4.46 | 73.58±1.29 | 93.02±1.83 |
| | Mahalanobis | 62.38±2.08 | 94.46±0.50 | 98.75±0.59 | 57.62±4.13 | 90.14±1.36 | 97.62±1.77 |
| | Energy Score | 47.47±3.33 | 85.47±1.73 | 95.58±2.48 | 20.38±4.88 | 76.46±3.08 | 93.68±3.77 |
| | GEM | **72.61**±3.21 | **94.59**±0.50 | **98.79**±0.61 | 57.40±7.44 | **90.17**±1.83 | 97.63±1.78 |
| | VOS | 53.55±3.11 | 86.81±0.73 | 95.72±0.62 | **59.08**±4.52 | 88.05±1.71 | **97.99**±1.92 |
| SVHN | Ours | **94.10** | **98.56** | **99.41** | **93.75** | **98.36** | **99.36** |
| | MSP | 52.41±5.19 | 92.11±0.62 | 98.32±0.61 | 15.66±4.09 | 71.37±2.43 | 92.89±2.46 |
| | Mahalanobis | 79.34±3.23 | 95.72±0.68 | 99.04±0.82 | 51.35±5.35 | 89.25±1.21 | 97.52±1.20 |
| | Energy Score | 64.20±2.44 | 91.05±1.06 | 97.66±1.99 | 14.59±2.79 | 74.10±1.73 | 93.65±1.85 |
| | GEM | 79±2.84 | 95.65±0.70 | 99.01±0.87 | 51.51±5.19 | 89.40±1.53 | 97.57±1.34 |
| | VOS | 75.41±4.21 | 95.26±0.71 | 98.99±0.83 | 51.66±4.03 | 87.13±2.21 | 96.85±1.96 |
| LSUN-Crop | Ours | 83.63 | 96.60 | 96.61 | 57.76 | 89.95 | 90.03 |
| | MSP | 69.07±3.87 | 95.64±0.50 | 99.13±0.51 | 33.44±5.09 | 83.71±1.81 | 96.32±1.72 |
| | Mahalanobis | 30.06±3.66 | 86.15±0.41 | 97.05±0.48 | 1.53±1.47 | 58.48±0.98 | 89.73±2.03 |
| | Energy Score | 91.89±2.74 | 98.40±0.24 | **99.67**±0.25 | 64.01±7.17 | 93.41±1.13 | 98.59±1.11 |
| | GEM | 30.20±3.70 | 86.09±0.34 | 97.03±0.39 | 1.70±0.77 | 58.42±1.52 | 89.70±2.11 |
| | VOS | **92.45**±3.54 | **98.44**±0.39 | **99.67**±0.47 | **91.94**±5.07 | **98.38**±1.69 | **99.65**±1.83 |
| LSUN-Resize | Ours | 85.41 | 96.84 | 96.86 | **88.49** | **97.56** | 97.55 |
| | MSP | 47.45±5.84 | 91.30±1.06 | 98.11±1.06 | 16.54±3.65 | 75.32±2.06 | 94.03±2.50 |
| | Mahalanobis | 35.64±3.59 | 88.12±0.63 | 97.45±0.77 | 67.20±5.62 | 93.97±1.06 | **98.70**±0.98 |
| | Energy Score | 71.75±6.18 | 94.12±1.14 | 98.64±1.44 | 21.38±4.53 | 79.29±1.68 | 94.97±1.92 |
| | GEM | 35.45±5.84 | 88.09±0.81 | 97.43±0.83 | 67.09±4.69 | 94.01±1.33 | **98.70**±1.30 |
| | VOS | **85.85**±5.83 | **97.19**±1.04 | **99.41**±1.17 | 63.91±3.72 | 90.95±1.97 | 97.74±2.24 |
| iSUN | Ours | 79.62 | 95.64 | 95.68 | **82.15** | **96.05** | 96.24 |
| | MSP | 43.40±3.54 | 89.72±1.21 | 97.72±1.26 | 17.02±4.12 | 75.87±2.05 | 94.20±1.79 |
| | Mahalanobis | 26.77±4.62 | 87.87±1.13 | 97.33±1.20 | 64.07±3.96 | 92.69±0.95 | **98.32**±0.88 |
| | Energy Score | 66.27±3.11 | 92.56±1.41 | 98.25±1.51 | 19.20±4.15 | 78.98±1.71 | 94.90±2.09 |
| | GEM | 36.80±7.29 | 87.85±1.25 | 93.33±1.21 | 64.10±5.15 | 92.73±1.03 | **98.32**±1.05 |
| | VOS | **82.29**±4.43 | **96.53**±1.27 | **99.25**±1.33 | 63.01±4.63 | 90.91±1.91 | 97.95±1.84 |
| Average Performance | Ours | **81.39** | **96.08** | 96.17 | **72.93** | **93.53** | 93.69 |
| | MSP | 50.63 | 91.46 | 98.07 | 19.87 | 75.97 | 94.09 |
| | Mahalanobis | 46.83 | 90.46 | 97.92 | 48.35 | 84.90 | 96.37 |
| | Energy Score | 68.31 | 92.32 | 97.96 | 27.91 | 80.44 | 95.15 |
| | GEM | 50.81 | 90.45 | 97.91 | 48.36 | 84.94 | 96.38 |
| | VOS | 77.91 | 94.84 | **98.60** | 65.92 | 91.08 | **98.03** |

approach with outlier exposure (Hendrycks et al., 2019). The average performance of (Hendrycks et al., 2019) is 95.6%, while ours stands at 96.2%. Therefore, our approach is comparable to the outlier exposure method.

**Large-Scale Datasets**: We evaluated our approach on large-scale datasets following a similar procedure. Specifically, we used the pretrained model CLIP (Radford et al., 2021) as the feature extractor, with ImageNet-1K (Huang et al., 2021) as the ID dataset and iNaturalist (Van Horn et al., 2018), SUN (Xiao et al., 2010), Textures (Cimpoi et al., 2014), and Places (Zhou et al., 2017) as the OOD datasets. The extracted text

Table 6: Comparison in computation time and memory cost during inference.

| Method | CIFAR-10 | | CIFAR-100 | |
|---|---|---|---|---|
| | Time/Sample | Memory | Time/Sample | Memory |
| Ours | 3.0ms | **1048.22MiB** | 3.0ms | **1134.32MiB** |
| MSP | **0.02ms** | 1825.21MiB | **0.02ms** | 1825.98MiB |
| Mahala. | 30.61ms | 1983.17MiB | 56.24ms | 1983.81MiB |
| Energy | 0.22ms | 1830.01MiB | 0.21ms | 1838.23MiB |
| GEM | 25.62ms | 1983.51MiB | 56.27ms | 1984.77MiB |
| VOS | 19.56ms | 1984.05MiB | 20.32ms | 1984.91MiB |

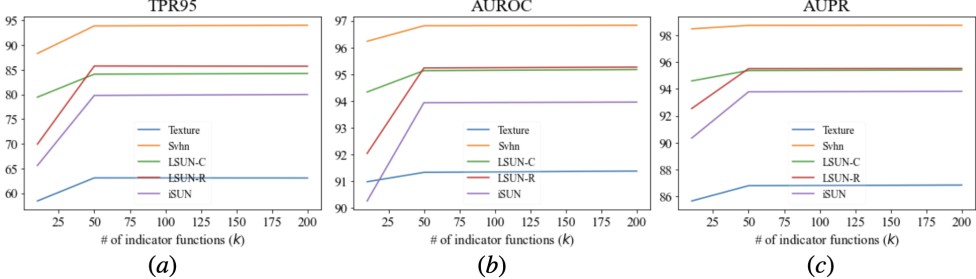

Figure 5: Performance of our approach with different numbers ($k = 10, 50, 200$) of condition functions for CIFAR-10 being ID data.

embeddings of ImageNet-1K labels served as the ID features for our OI-based detector. The utilized prompt is "the $< >$" with the blank filled by specific labels. For example, if the class is cat, then the generated prompt would be "the cat". During inference, for a given image, we extracted its image embedding using CLIP and calculated the OI value between this embedding and the text embeddings of ID labels. To build the contaminated dataset, we merged all ID and OOD features, then randomly selected 100 features. During testing, each feature was subtracted from the mean vector of this contaminated dataset. The compared methods are allowed to use ImageNet-1K training data to train or fine-tune their OOD detectors. The results, shown in Table 7, demonstrate consistent performance of our approach on large-scale datasets. We have also compared our approach with the more recent method DCM (Choi et al., 2024), which minimizes confidence on an uncertainty dataset. DCM achieves an overall AUROC performance of 97.4%, compared to 93.89% for our method. Although DCM delivers slightly better OOD detection results, our approach demonstrates several advantages in terms of training time efficiency, memory cost, and dataset requirements. For example, DCM requires access to an uncertainty dataset containing both ID and OOD data for model fine-tuning, whereas our approach only requires the mean of the uncertainty dataset, offering much faster training times and lower memory costs, making it less restrictive.

**Image Corruptions**: We tested our approach using ImageNet-1K as the ID dataset and corrupted images as the OOD dataset (Hendrycks & Dietterich, 2019). Specifically, we have tested our approach on one of the ImageNet-C datasets, specifically the image blur dataset. We note that our approach performs well on image corruption tasks, including defocus blur, glass blur, and motion blur, achieving AUROCs of 98.92%, 98.37%, and 97.93%, respectively.

**Gaussian-Based Approaches**: We feed raw data into WideResNet pretrained models to extract complex high-dimensional features whose shape is $128 \times 8 \times 8$. The pretrained models are downloaded from (Morteza & Li, 2022). The compared methods are MSP (Hendrycks & Gimpel, 2017), Mahalanobis (Lee et al., 2018), Energy score (Liu et al., 2020), GEM (Morteza & Li, 2022), and VOS (Du et al., 2022). All baseline methods are evaluated in the same feature space with their optimal hyperparameters for fair comparisons. We formed a small dataset by randomly selecting ten samples from each class to form the available ID dataset. Table 5 shows the experimental results of our approach compared with Gaussian-based approaches for the feature-level OOD detection when CIFAR-10 and CIFAR-100 are the ID datasets and Textures, SVHN, LSUN-Crop,

Table 7: AUROC on large-scale datasets. The ID dataset is ImageNet-1K. The backbone is ViT-B/16.

| OOD Datasts | iNaturalist | SUN | Places | Textures | Ave. |
|---|---|---|---|---|---|
| MSP (Hendrycks & Gimpel, 2017) | 87.44 | 79.73 | 79.67 | 79.69 | 81.633 |
| ODIN (Liang et al., 2018) | 94.65 | 87.17 | 85.54 | 87.85 | 88.803 |
| GradNorm (Huang et al., 2021) | 72.56 | 72.86 | 73.70 | 70.26 | 72.345 |
| ViM (Wang et al., 2022b) | 93.16 | 87.19 | 83.75 | 87.18 | 87.820 |
| KNN (Sun et al., 2022) | 94.52 | 92.67 | 91.02 | 85.67 | 90.970 |
| VOS (Du et al., 2022) | 94.62 | 92.57 | 91.23 | 86.33 | 91.188 |
| NPOS (Tao et al., 2022) | 96.19 | 90.44 | **89.44** | 88.80 | 91.218 |
| Ours | **99.57** | **92.78** | 88.31 | **94.92** | **93.895** |

Table 8: Average performance of different OOD detectors for backdoor detection.

| Metrics (%) | Ours | STRIP | Mahalanobis | GEM | MSP |
|---|---|---|---|---|---|
| TPR95 | 89.40 | 39.60 | 56.97 | **91.57** | 39.24 |
| AUROC | **96.68** | 70.30 | 75.94 | 58.08 | 54.92 |
| AUPR | **95.42** | 68.76 | 76.37 | 75.88 | 60.52 |

LSUN-Resize, and iSUN are the OOD datasets. Table 6 shows the computation time and memory cost of our approach and the other methods. Our approach outperforms the other methods using the least memory, showing the highest TPR95 and AUROC on average. The AUPR of our approach is in the same range as other baseline methods. We further evaluated our approach using different numbers $k$ of conditions functions and plotted the results in Fig. 5. From the figure, the performance of our approach increases with $k$ and eventually converges, which is consistent with Fig. 3(b).

**Backdoor Detection**: We applied our approach to backdoor detection (Gu et al., 2019) by considering clean samples as ID data and poisoned samples as OOD data. We created different backdoored models and used them to extract data features. We compare with previous baseline methods plus STRIP (Gao et al., 2019). The average performance is given in Table 8. Our approach detects various attacks (Chen et al., 2017; Liu et al., 2019; Li et al., 2021b; Fu et al., 2022; Nguyen & Tran, 2021) and outperforms other baseline methods by showing the highest average AUROC. Details are provided in Appendix H.

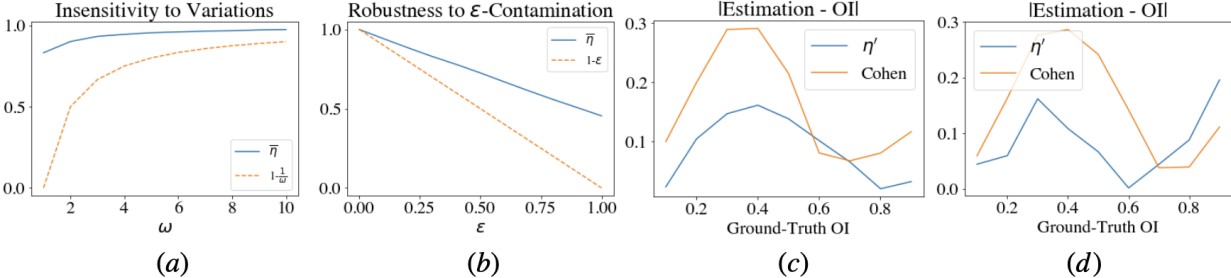

Figure 6: (a): illustration of **Proposition 5.1**. (b): illustration of **Proposition 5.2**. (c,d): the absolute value of estimation errors between the ground-truth OI and $\overline{\eta}'$ for uniform distributions (c) and truncated Gaussian distributions (d).

## 5 Analysis

### 5.1 Mathematical Properties

**Proposition 5.1.** *Consider uniform distribution for $D^+$ in $[0,1]$ and $1 + \sin 2\pi\omega x$ for $D^-$ in $[0,1]$, with $g(x) = \mathbb{1}\{||x|| \leq r\}$, the proposed confidence score function $\overline{\eta}$ is insensitive to small distribution variations:*

$$\overline{\eta}(D^+, D^-) \geq 1 - \frac{1}{\omega} \quad \forall \omega \geq 1. \tag{7}$$

**Proposition 5.2.** *Let $\epsilon \in [0,1]$, then for arbitrary $D^+$, $D^-$, and $g$, the proposed confidence score function $\overline{\eta}$ is robust against Huber-$\epsilon$ contamination:*

$$\overline{\eta}(D^+, (1-\epsilon)D^+ + \epsilon D^-) \geq 1 - \epsilon. \tag{8}$$

The insensitivity and robustness are illustrated in Fig. 6(a,b). The following theorem shows the potential of our OI-based confidence score function for estimating the model accuracy.

**Theorem 5.3.** *Assume that $D$ and $D^*$ are two different data distributions with $\eta(D, D^*) < 1$ and denote the overall accuracy of the model on $D^*$ as Acc. If a model has a training accuracy $p$ on $D$ and a testing accuracy $q$ on $D^* \setminus D$, then we have*

$$Acc \leq (p - q)(1 - \frac{1}{2r_B}||\mu_D - \mu_{D^*}|| - \max_g \frac{r_B - r_{A(g)}}{2r_B} |\mathbb{E}_D[g] - \mathbb{E}_{D^*}[g]|) + q. \tag{9}$$

**Theorem 5.3** provides a theoretical interest for OOD generalization and is also empirically useful when the domain shift happens in the backdoor setting, where the model has zero clean accuracy on poisoned data (i.e., $q = 0$). Define the clean distribution as $D$, poisoned distribution as $D^p$, and a testing distribution $D^*$ with $D^* = \sigma D + (1 - \sigma)D^p$, where $\sigma \in [0,1]$. Then (9) becomes

$$Acc \leq p(1 - \frac{1 - \sigma}{2r_B}||\mu_D - \mu_{D^p}|| - (1 - \sigma) \max_g \frac{r_B - r_{A(g)}}{2r_B} |\mathbb{E}_D[g] - \mathbb{E}_{D^p}[g]|). \tag{10}$$

The experimental illustration is given in Appendix I.

### 5.2 OI Estimation

Let $\overline{\eta}' = \max\{0, 1 - \max_g \frac{r_B - r_{A(g)}}{2r'} |\mathbb{E}_{D^+}[g] - \mathbb{E}_{D^-}[g]| - \frac{1}{2r'}||\mu_{D^+} - \mu_{D^-}||\}$ be the variant of $\overline{\eta}$ given in equation 6, where $r' \leq r_B$. For any given $r'$, we could use Alg. 1 to calculate $\overline{\eta}'$ as an estimate of OI. To validate $\overline{\eta}'$, we consider estimate OI in $\mathbb{R}^4$. We set $r'$ as the median Euclidean norm among all vectors in $B$, use indicator functions $g_j(x) = \mathbb{1}\{||x|| \leq r_j\}$, and choose $k = 100$ and $m = 50$. The utilization distributions are truncated Gaussian and uniform. We merged the two data clusterings and computed the mean vector of the combined set. We set this mean vector as the origin. The results are illustrated in Fig. 6(c,d). We empirically observed that the computation and memory costs of using the kernel estimator (Pastore & Calcagnì, 2019) in $\mathbb{R}^4$ are much higher than ours and Cohen's d measure (Inman & Bradley Jr, 1989). Therefore, we only show the comparison results with Cohen's d measure. The findings suggest that our OI-based confidence score function shows promise for applications in OI estimation.

## 6 Discussion

### 6.1 Joint Selective Classification-OOD Detection

This work focuses on OOD detection. However, recent advances in selective classification (Narasimhan et al., 2024) closely overlap with OOD detection, as it is crucial to determine when the model should make a prediction. ID samples on which the model is not confident should also be taken into account. Exploring whether our approach can be applied to selective classification is part of our future work.

## 6.2 Limitation and Broader Impact

Zhang et al. (2021a) proves that for any OOD detector, there exist situations where the OOD samples lie in the high probability or density regions of ID samples, and the OOD method performs no better than the random guess. We empirically observed that by shifting in- and out-distribution clusterings to the origin, our approach becomes less effective, supporting Zhang et al. (2021a)'s claim. For example, in the hard-OOD detection task where CIFAR-10 is the ID dataset and CIFAR-100 is the OOD dataset, the AUROC of our approach decreases to around 80%. Due to this limitation, the attacker may use our work to design stealthy OOD samples to evade our detection, causing machine learning systems to malfunction. Nevertheless, we believe that the positive impact of our approach far outweighs any potential negative societal impact.

## 7 Conclusion

This paper proposes a novel OOD detection approach that strikes a favorable balance between accuracy and computational efficiency. The utilized OI-based confidence score function is non-parametric, computationally lightweight, and demonstrates effective performance. Empirical evaluations indicate that the proposed OI-based OOD detector is competitive with state-of-the-art OOD detectors across a variety of scenarios, while maintaining a more frugal computational and memory footprint. The proposed OI-based confidence score function shows an insensitivity to small distributional shifts and a robustness against Huber $\epsilon$-contamination. Additionally, it shows potential for estimating OI and model accuracy in specific applications. Overall, this paper showcases the effectiveness of using the OI-based metric for OOD detection.

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

## A  Proof of Theorem 3.3

**Definition A.1** (Total Variation Distance (TVD))**.** TVD $\delta : \mathbb{R}^n \times \mathbb{R}^n \to [0, 1]$ of $P$ and $Q$ is defined as:

$$\delta(P, Q) = \frac{1}{2} \int_{\mathbb{R}^n} |f_P(x) - f_Q(x)| \, dx. \tag{11}$$

Since OI + TVD = 1, we have $\eta = 1 - \delta = 1 - \delta_A - \delta_{\mathbb{R}^n \setminus A}$.

*Proof.* Let $f_{D^+}$ and $f_{D^-}$ be the probability density functions for $D^+$ and $D^-$. From **Definition** A.1, we have

$$\eta(D^+, D^-) = 1 - \delta_A(D^+, D^-) - \delta_{A^c}(D^+, D^-). \tag{12}$$

Using (12), triangular inequality, and boundedness, we obtain

$$||\mu_{D^+} - \mu_{D^-}|| = ||\int_B x \left(f_{D^+}(x) - f_{D^-}(x)\right) dx|| \tag{13}$$

$$\leq \int_B ||x(f_{D^+}(x) - f_{D^-}(x))|| dx \tag{14}$$

$$= \int_A ||x|| \cdot |f_{D^+}(x) - f_{D^-}(x)| dx \tag{15}$$

$$+ \int_{A^c} ||x|| \cdot |f_{D^+}(x) - f_{D^-}(x)| dx \tag{16}$$

$$\leq 2r_A \delta_A + 2r_{A^c} \delta_{A^c} \tag{17}$$

$$= 2r_A \delta_A + 2r_{A^c} (1 - \delta_A - \eta(D^+, D^-)) \tag{18}$$

which implies (3). Since $1 - \delta_A - \eta(D^+, D^-) \geq 0$, we can replace $r_{A^c}$ with $r_B$ in (18) to get (4). $\qquad\square$

## B    Proof of Corollary 3.4

*Proof.* Let $g : B \to \{0, 1\}$ be a condition function and define $A(g) = \{x \mid g(x) = 1, x \in B\}$. According to the definition of $\delta_A$ and triangular inequality, we have

$$\delta_{A(g)}(D^+, D^-) = \frac{1}{2} \int_{A(g)} |f_{D^+}(x) - f_{D^-}(x)| dx \tag{19}$$

$$\geq \frac{1}{2} |\int_{A(g)} f_{D^+}(x) - f_{D^-}(x) dx| \tag{20}$$

$$= \frac{1}{2} |\int_{\mathbb{R}^n} g(x) f_{D^+}(x) - g(x) f_{D^-}(x) dx|$$

$$= \frac{1}{2} |\mathbb{E}_{D^+}[g] - \mathbb{E}_{D^-}[g]| . \tag{21}$$

Applying (21) into **Theorem** 3.3 gives **Corollary** 3.4. $\qquad\square$

## C    Proof of Proposition 5.1

*Proof.* Since the support is $[0, 1]$, we have $r_B = 1$, $\mu_{D^+} - \mu_{D^-} = -\int_0^1 x \sin 2\pi\omega x dx \leq \frac{1}{\omega}$, $\mathbb{E}_{D^+}[g] - \mathbb{E}_{D^-}[g] = -\int_0^1 g(x) \sin 2\pi\omega x dx \leq \frac{1}{\omega}$, and $\overline{\eta} \geq 1 - \frac{1}{2\omega} - \max_g \frac{1 - r_{A(g)}}{2} \frac{1}{\omega} \geq 1 - \frac{1}{\omega}$. $\qquad\square$

## D    Proof of Proposition 5.2

*Proof.* Denote $Q = (1 - \epsilon)D^+ + \epsilon D^-$, then $\mu_{D^+} - \mu_Q = \epsilon(\mu_{D^+} - \mu_{D^-})$, $\mathbb{E}_{D^+}[g] - \mathbb{E}_Q[g] = \epsilon(\mathbb{E}_{D^+}[g] - \mathbb{E}_{D^-}[g])$, and $\overline{\eta}(D^+, Q) = 1 - \epsilon(\frac{1}{2r_B}||\mu_{D^+} - \mu_{D^-}|| + \max_g \frac{r_B - r_{A(g)}}{2r_B} |\mathbb{E}_{D^+}[g] - \mathbb{E}_{D^-}[g]|) \geq 1 - \epsilon$. $\qquad\square$

## E    Proof of Theorem 5.3

*Proof.* Let $f_D$ and $f_{D^*}$ be their probability density functions, $Acc = \int_{x \sim D^*} \left(p \frac{\min\{f_D(x), f_{D^*}(x)\}}{f_{D^*}(x)} + q \left(1 - \frac{\min\{f_D(x), f_{D^*}(x)\}}{f_{D^*}(x)}\right)\right) \times f_{D^*}(x) dx = p\eta(D, D^*) + q(1 - \eta(D, D^*)) \leq (p - q)(1 - \frac{1}{2r_B}||\mu_D - \mu_{D^*}|| - \max_g \frac{r_B - r_{A(g)}}{2r_B} |\mathbb{E}_D[g] - \mathbb{E}_{D^*}[g]|) + q$. $\qquad\square$

## F    Details of Used Datasets

Details of used datasets are given in Table 9.

Table 9: Information on other utilized datasets.

| Dataset | Dimension | # Class | # Training | # Testing |
|---|---|---|---|---|
| CIFAR-10 | $3 \times 32 \times 32$ | 10 | 50000 | 10000 |
| CIFAR-100 | $3 \times 32 \times 32$ | 100 | 50000 | 10000 |
| Textures | $3 \times 32 \times 32$ | 47 | N/A | 5640 |
| SVHN | $3 \times 32 \times 32$ | 10 | N/A | 26032 |
| LSUN_C | $3 \times 36 \times 36$ | 1 | N/A | 10000 |
| LSUN_R | $3 \times 32 \times 32$ | 1 | N/A | 10000 |
| iSUN | $3 \times 32 \times 32$ | 1 | N/A | 8925 |
| MNIST | $1 \times 28 \times 28$ | 10 | 60000 | 10000 |
| GTSRB | $3 \times 32 \times 32$ | 43 | 39209 | 12630 |
| YouTube Face | $3 \times 55 \times 47$ | 1283 | 103923 | 12830 |
| sub-ImageNet | $3 \times 224 \times 224$ | 200 | 100000 | 2000 |

## G   Details for Fig. 2

The numerical results are given in Table 10.

Table 10: AUROC (%) for different methods on UCI datasets.

| Ours | L1-Ball | K-Center | Parzen |
|---|---|---|---|
| $97.67 \pm 2.55$ | $90.19 \pm 22.21$ | $92.4 \pm 10.24$ | $93.99 \pm 9.92$ |
| Gaussian | K-Mean | 1-Nearest Neighbor | K-Nearest Neighbor |
| $95.83 \pm 6.45$ | $94.51 \pm 5.7$ | $93.24 \pm 11.05$ | $93.24 \pm 11.05$ |
| Auto-Encoder Network | Linear Programming | Principal Component | Lof Range |
| $87.81 \pm 22.15$ | $81.5 \pm 34.42$ | $81.8 \pm 25.18$ | $86.04 \pm 23.54$ |
| Nearest Neighbor Distance | Minimum Spanning Tree | Minimum Covariance Determinant | Self Organizing Map |
| $84.54 \pm 15.91$ | $92.32 \pm 11.4$ | $94.1 \pm 11.56$ | $93.6 \pm 7.49$ |
| Support Vector Machine | Minimax Probability Machine | Mixture Gaussians | Local Outlier Factor |
| $78.69 \pm 36.72$ | $77.84 \pm 35.57$ | $93.61 \pm 11.15$ | $89.04 \pm 13.77$ |
| Naive Parzen | Local Correlation Integral | | |
| $97.41 \pm 2.59$ | $96.79 \pm 2.45$ | | |

## H   Backdoor Detection

The datasets are balanced by having an equal number of clean and poisoned samples. For each backdoor attack, we assume that a small clean validation dataset is available (i.e., ten samples from each class) at the beginning. Therefore, the poisoned samples (i.e., samples attached with triggers) can be considered OOD, whereas the clean samples can be considered ID. The metrics used are: TPR95 (i.e., the detection accuracy for poisoned samples when the detection accuracy for clean samples is 95%), AUROC, and AUPR. We have carefully fine-tuned the baseline methods' hyperparameters to ensure their best performance over other hyperparameter choices. Fig. 7 shows the utilized triggers, and Table 11 shows the details for backdoor detection. For most triggers, our method has over 96% of TPR95, over 97% of AUROC, and 95% of AUPR. Our detector is robust against the latest or advanced backdoor attacks, such as Wanet, invisible trigger, all

Table 11: Comparison results for backdoor detection (higher number implies higher accuracy).

| Datasets | Trigger | Metrics (%) | Ours | STRIP | Mahalanobis | GEM | MSP |
|---|---|---|---|---|---|---|---|
| MNIST | All label | TPR95 | 83.05 | 2.58 | 50.83 | **100** | 100 |
| | | AUROC | **96.13** | 44.69 | 90.78 | 50.43 | 50 |
| | | AUPR | **94.20** | 35.47 | 86.71 | 70.94 | 70.83 |
| MNIST | Naive.1 | TPR95 | **100** | 98.85 | 99.86 | 100 | 5.11 |
| | | AUROC | **97.50** | 97.32 | 97.49 | 53.95 | 51.64 |
| | | AUPR | 96.17 | 95.95 | **96.38** | 74.74 | 50.41 |
| MNIST | Naive.2 | TPR95 | 96.53 | 67.46 | 35.16 | **100** | 14.69 |
| | | AUROC | **97.28** | 93.67 | 78.63 | 53.51 | 58.14 |
| | | AUPR | **95.75** | 89.85 | 78.65 | 74.62 | 64.16 |
| CIFAR-10 | TCA.1 | TPR95 | **100** | 35.68 | 100 | 100 | 4.38 |
| | | AUROC | **97.50** | 83.00 | 97.49 | 50 | 49.23 |
| | | AUPR | 95.47 | 73.22 | **97.84** | 76.32 | 52.64 |
| CIFAR-10 | TCA.2 | TPR95 | **100** | 27.86 | 100 | 100 | 0.02 |
| | | AUROC | **97.50** | 68.79 | 97.49 | 50 | 29.90 |
| | | AUPR | **97.63** | 72.41 | 95.86 | 67.86 | 18.05 |
| CIFAR-10 | Wanet | TPR95 | 37.87 | 0.07 | 20.35 | 22.90 | **100** |
| | | AUROC | **92.74** | 34.97 | 50.61 | 57.81 | 50 |
| | | AUPR | **89.95** | 37.42 | 57.30 | 68.48 | 74.87 |
| GTSRB | Moving | TPR95 | **99.99** | 54 | | | |
| | | AUROC | **85.39** | 7.29 | Fail: dependent data features | | |
| | | AUPR | **96.96** | 89.07 | | | |
| GTSRB | Filter | TPR95 | **85.39** | 7.29 | | | |
| | | AUROC | **96.54** | 38.92 | Fail: dependent data features | | |
| | | AUPR | **95.42** | 38.81 | | | |
| GTSRB | Wanet | TPR95 | **100** | 1.24 | 0.51 | 100 | 100 |
| | | AUROC | **97.50** | 36.31 | 54.46 | 50 | 50 |
| | | AUPR | **97.62** | 39.53 | 48.92 | 75.23 | 75.23 |
| YouTube Face | Sunglasses | TPR95 | 73.37 | 83.03 | 71.64 | **98.58** | 13.06 |
| | | AUROC | **95.21** | 94.80 | 94.38 | 84.29 | 66.55 |
| | | AUPR | 93.00 | **95.54** | 94.63 | 88.83 | 53.27 |
| YouTube Face | Lipstick | TPR95 | **96.64** | 90.14 | 90.88 | 94.18 | 3.73 |
| | | AUROC | **97.21** | 93.15 | 93.26 | 80.80 | 50.14 |
| | | AUPR | **96.30** | 94.98 | 95.09 | 86.53 | 53.27 |
| sub-ImageNet | Invisible | TPR95 | **100** | 7.01 | 0.5 | 100 | 51.40 |
| | | AUROC | **97.49** | 66.26 | 4.78 | 50 | 93.61 |
| | | AUPR | **96.53** | 62.83 | 12.27 | 75.26 | 92.46 |
| Average Performance | | TPR95 | 89.40 | 39.60 | 56.97 | **91.57** | 39.24 |
| | | AUROC | **96.68** | 70.30 | 75.94 | 58.08 | 54.92 |
| | | AUPR | **95.42** | 68.76 | 76.37 | 75.88 | 60.52 |

Figure 7: Pictures under "Triggers" are poisoned samples. Pictures under "Clean" are clean samples.

label attack, and filter attack, whereas the baseline methods show low performance on those attacks. The Gaussian-based baseline methods encountered an error for two cases because the data features are dependent.

## I Experimental Illustration of Theorem 5.3

**Setup**: We use MNIST, GTSRB, YouTube Face, and sub-ImageNet and their domain-shifted versions given in Fig. 8 to illustrate the theorem. We set $\sigma = 0, 0.1, ..., 0.9, 1$ and calculated the RHS of (10) using $L_1$, $L_2$, and $L_\infty$ norms in the raw image input space, model output space, and hidden layer space. We use $g_j(x) = \mathbb{1}\{||x|| \le r_j\}$. Fig. 9 shows the model accuracy and corresponding upper bounds with different norms and $\sigma$s. The model accuracy is below all calculated upper limits, validating the theorem. The difference between model accuracy and the calculated upper bound accuracy can reflect the extent of the domain shift in the test dataset. From Fig. 9, a large difference reflects a large domain shift. When the domain shift vanishes, the model accuracy and the calculated upper bound accuracy are close.

From Fig. 9, the calculated upper bound accuracy varies with spaces. The inference is that a low calculated upper bound accuracy implies a high likelihood of detecting domain-shifted data in that particular utilized space. For example, in Fig. 9 MNIST and YouTube, the input space with $L_\infty$ norm shows the lowest upper bound accuracy. Therefore, domain-shifted samples will likely be detected in the input space. Indeed, from Fig. 8, even vision inspection can easily detect them. As for GTSRB and ImageNet, the input space has the highest upper limit lines. Therefore, domain-shifted samples are less likely to be detected in the input space. Fig. 8 shows that the visual inspection can barely detect them. However, the hidden layer space gives the lowest upper bound accuracy. Therefore, domain-shifted samples are more likely to be distinguished in the hidden layer space, as shown in Fig. 8. The AUROC is 96.54% for GTSRB and 97.49% for ImageNet according to Table 11.

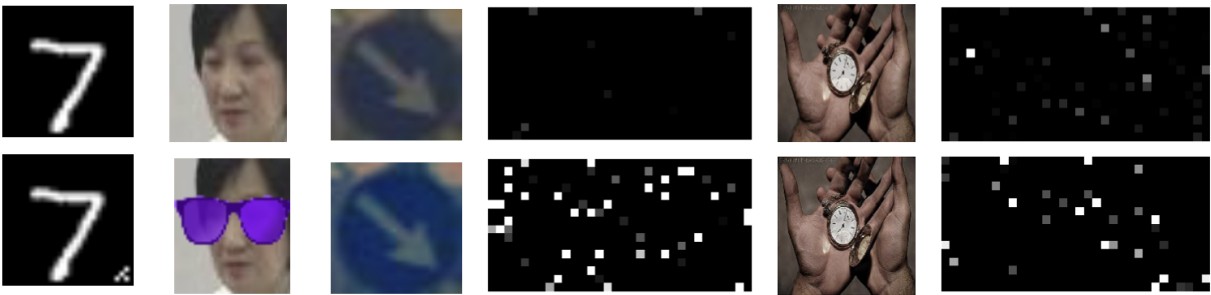

Figure 8: Top: original samples. Bottom: domain-shifted samples. From left to right: MNIST, YouTube Face, GTSRB, GTSRB in hidden layer space, ImageNet, ImageNet in hidden layer space.

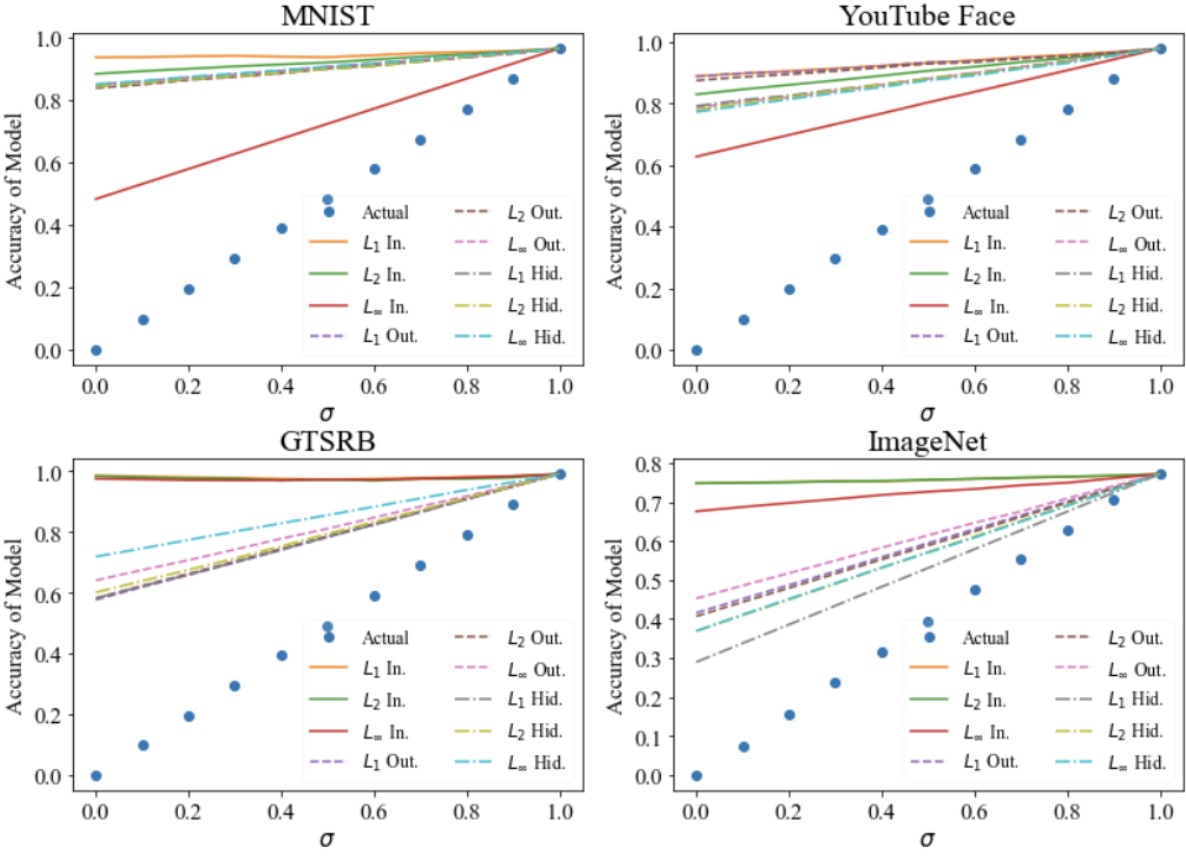

Figure 9: The model accuracy vs. equation 10 calculated with $L_{1,2,\infty}$ norms in input, output, and hidden spaces. X-axis is the ratio of clean samples to the entire testing samples.

