# OpenReview forum: "Out-of-Distribution Detection with Overlap Index"
_TMLR — Rejected by TMLR_

### Review · Reviewer_KgHs · 2025-04-01

**Summary Of Contributions:**

This paper proposes an OOD detection method that uses the overlap index (OI), which is essentially 1 minus the TV distance. The authors derive an upper bound for OI, and propose to use that upper bound as OOD detection. The upper bound $\bar{\eta}$ is larger on ID data than OOD data. The authors test their method on some real datasets.

**Audience:**

Yes

**Claims And Evidence:**

No

**Requested Changes:**

1. Elaborate on the details of your method. How is $D^-$ chosen? What if ID and OOD data is mixed? Why is $A(g)$ necessary? Why do you choose $g$ in this way?
2. A large number of OOD detection methods have been proposed in recent years. The newest methods compared in this paper are from 2022, which I am not sure is the state of the art.

**Strengths And Weaknesses:**

## Strengths
1. Overall the paper is easy to read.
2. From Figure 1(a) it does seem that the metric is different on ID and OOD, though how this figure is obtained is unclear to me.

## Weaknesses
1. The description of the method is confusing
  - It seems that the method needs a set of samples from ID $D^+$ and a set of samples from OOD $D^-$. Thus, it is confusing to me how it can detect whether each individual sample is OOD or not.
  - I am not sure if the method would still work if $D^-$ consists of both ID and OOD data. After all, the OOD detection problem aims to tell ID and OOD apart, so this is the common scenario.
  - I am not sure how Figure 1(a) is obtained. How do you choose the $D^-$ in Figure 1(a)? Is it a set of samples from one of the 9 OOD classes?
  - In Corollary 3.4, I do not understand why you use $r_{A(g)}$. If $0 \in B$, then the argmax of $A$ in Eqn. (5) is simply $\{0\}$, and $r_A = 0$. I cannot see why this $A(g)$ is necessary.
  - In Corollary 3.4, I believe it should be $| E_{D^+}[g] - E_{D^-}[g] |$, with the absolute value.

Since the method itself is unclear, I don't think the claim of this paper is sufficiently supported.

2. The authors claim that the bound in Corollary 3.4 is novel. To me, this bound is a simple derivation from the definition. There does not seem to be any technical difficulty. In addition, since OI is 1 minus TV distance, and there are lots of bounds for the TV distance, I am not quite sure if this bound is novel, though I cannot point to a specific reference.

3. The experimental results are not really impressive. In Table 5, the proposed method is only the best on SVHN. Although the method has the highest average performance, this is largely because its performance on SVHN is much higher than other compared methods (though I cannot see why this should be the case). If SVHN is neglected then the proposed method is not the best.

---

> ### Author Response · Authors · 2025-05-11
> **Weakness 1**
>
> 1. Our approach requires only a small dataset of pure ID data to detect OOD samples. It does not need any OOD samples for OOD detection. Please refer to the second paragraph in the “**Choice of g**” section in Sec. 3.3 of our original paper for a detailed explanation of how it detects OOD samples. The key idea is that our approach calculates the proposed OI upper bound between an individual input and the small ID dataset. If the calculated bound exceeds a pre-defined threshold, the input is classified as ID; otherwise, it is considered OOD. We have highlighted the relevant paragraph with the title “**The OI-Based Confidence Score Function**” for further clarification.
>
> 2. Yes, our approach functions by assigning an OOD score to each individual sample, as explained in our response to your first concern. Therefore, it remains effective even when  $D^-$  contains both ID and OOD data.
>
> 3. $D^+$  consists of all samples from the “plane” class, while  $D^-$  includes all samples from the remaining nine classes. We have further clarified this in the caption.
>
> 4. The utilization of  $r_A(g)$  helps further separate the ID and OOD distributions, as demonstrated by comparing Figure 1(a) and Figure 1(b). In Figure 1(b), where  $r_A(g)$  is not used, the scores between ID and OOD are less separable compared to Figure 1(a), which incorporates  $r_A(g)$. This comparison is highlighted in our original manuscript in Sec. 3.4, “**Visualization**.”
>
> 5. Yes, it is the absolute value and is shown in our original manuscript. We have further clarified this in corollary 3.4 in the updated version.

---

> ### Author Response · Authors · 2025-05-11
> **Weaknesses 2, 3, and Requested Changes**
>
> **W2**
>
> Our bound is novel in the following aspects:
>
> 1. To the best of our knowledge, our bound is not mentioned in any other references.
>
> 2. The bound possesses several key properties, such as not requiring the distributions to be identical, being insensitive to small distributional variations, and being robust to Huber $\epsilon$-contamination outliers.
>
> 3. It demonstrates usefulness in OOD detection, as shown in the experimental section of our manuscript. For instance, the OOD detector based on our bound is lightweight, non-parametric, and effective on large, high-dimensional datasets.
>
> **W3**
>
> In Sec. 4, our approach demonstrates strong overall performance across multiple tasks, evaluated using various metrics such as AUROC, AUPR, time complexity, and memory complexity. Specifically, in Table 5, our approach achieves either the best or second-best performance in most cases. Please also refer to our response to **Reviewer QcNM W3**, where we show the effectiveness of our approach on image corruption tasks. Based on all the experimental results, we believe that our approach exhibits excellent performance and significantly outperforms prior methods.
>
> **Requested Changes**
>
> 1. Please refer to our response to your **Weakness 1**. Regarding the choice of  g , we have discussed this in our original manuscript in Sec. 3.3, under the paragraph titled “**Choice of  g**.” The way we selected  g  is well-regularized, computationally efficient, and empirically effective. However, as mentioned in our paper, the choice is not unique.
>
> 2. In the updated manuscript, we have also compared our approach with additional recent methods. Please refer to our response to **Reviewer QcNM Weakness 1** for further details.

---

### Review · Reviewer_QcNM · 2025-04-09

**Summary Of Contributions:**

The paper introduces a novel approach to identifying out-of-distribution (OOD) samples in machine learning models by leveraging the Overlap Index (OI) as a confidence score function. Overlapping index is equal to 1 - the total variation distance between two distributions, and it calculates how much overlap there is between two distributions P and Q.

Traditional OOD detection methods can be classified into two classes:

1. **Non-parameteric**: These methods are computationally cheap, but does not work well on large and complex high dimensional datasets.

2. **Deep learning**: These methods use trained neural networks in some ways. However, these methods can be very slow in certain cases.

This paper’s method aims to address common challenges in OOD detection, such as computational inefficiency and limited interpretability, by offering a non-parametric, lightweight, and interpretable solution.

**Audience:**

Yes

**Broader Impact Concerns:**

No such concerns

**Claims And Evidence:**

Yes

**Requested Changes:**

Please address the weaknesses above.

# References

[1] Dream the Impossible: Outlier Imagination with Diffusion Models, https://arxiv.org/abs/2309.13415

[2] Conservative Prediction via Data-Driven Confidence Minimization, https://arxiv.org/abs/2306.04974

[3] Exploring the Limits of Out-of-Distribution Detection, https://arxiv.org/abs/2106.03004

[4] A Simple Fix to Mahalanobis Distance for Improving Near-OOD Detection, https://arxiv.org/abs/2106.09022

[5] Benchmarking Neural Network Robustness to Common Corruptions and Perturbations, https://arxiv.org/abs/1903.12261

[6] Plugin estimators for selective classification with out-of-distribution detection, https://arxiv.org/abs/2301.12386

**Strengths And Weaknesses:**

## Strengths


1. **Overlap Index-Based Confidence Score**: The paper proposes using the Overlap Index to evaluate the likelihood that a given input belongs to the same distribution as in-distribution (ID) samples. I like the approach, as Algorithm 1 is simple to understand/implement.

2. **Theoretical results**: The authors derive an upper bound for the Overlap Index, which is utilized to obtain a practical confidence score function for OOD detection. The theoretical results, while not very sophisticated, are still useful.

3. **Empirical results**: The paper has pretty extensive experimental results.

## Weaknesses

1. Table 7 numbers are very strong, however comparisons with [1], [2] can make the paper stronger.

2. It is unclear to me to what extent the method is useful for near or hard OOD detection problems [3], [4]. For example, how well does the model perform when CIFAR-10 is used at the ID dataset and CIFAR-100 is used as the OOD dataset, or vice versa?

3. How well does the proposed model work against image corruptions? For example, can it detect ImageNet-C from ImageNet?

4. Finally, the most practical use-case for reliability is not detecting ID vs OOD detection, but rather finding when to make a prediction or not. Some discussion around the joint selective classification-OOD detection problem setup [6] can be useful.

---

> ### Author Response · Authors · 2025-05-11
> **Rebuttal**
>
> **W1**:
>
> ***Response***
>
> Thank you for referring to these works. We have included a comparison with [2] at the end of the “Large-Scale Datasets” section in Sec. 4.2. Specifically, we found that [2] achieves slightly better performance, with an AUROC of 97.4%, compared to 93.89% for our method. However, our approach offers several advantages in terms of training time efficiency, memory cost, and dataset requirements. For instance, DCM requires access to an uncertainty dataset containing both ID and OOD data for model fine-tuning, whereas our approach only requires the mean of the uncertainty dataset, resulting in much faster training times and lower memory costs, making it less restrictive.
>
> Regarding [1], we currently do not have a model available for ImageNet-1k, as the authors did not report test results on ImageNet-1k nor provide a pre-trained model for evaluation. Therefore, we would need to train their model on ImageNet-1k from scratch. As a result, we are unable to provide the numbers at this time. However, we have discussed their work in the related work section.
>
> ***Action***
>
> We have included the comparison with [2] in Sec. 4.2 and the discussion of [1] in Sec. 2.1.
>
> **W2**
>
> ***Response***
>
> We tested the performance of our approach using CIFAR-10 as the ID dataset and CIFAR-100 as the OOD dataset, and vice versa. The AUROC of our approach for the hard OOD detection problem is around 79.45% and 80.22%, respectively. As acknowledged in our original paper, when the OOD distribution is close to the ID distribution, our bound becomes looser, leading to a reduction in performance. We recognize this as a limitation, which we aim to address in future work. Nevertheless, we believe that our manuscript demonstrates the promising aspects of our approach, such as its non-parametric nature, lightweight design, and effectiveness for regular OOD detection, including image corruption tasks, which outweigh the identified limitation.
>
> ***Action***
>
> We have added the above discussion in the limitation section in Sec. 6.2.
>
> **W3**
>
> ***Response***
>
> Per your suggestion, we have tested our approach on one of the ImageNet-C datasets, specifically the image blur dataset. We note that our approach performs well on image corruption tasks, including defocus blur, glass blur, and motion blur, achieving AUROCs of 98.92%, 98.37%, and 97.93%, respectively.
>
> ***Action***
>
> We have included the above results on image corruptions in Sec. 4.2.
>
> **W4**
>
> Thank you for your suggestion. We have discussed the selective classification-OOD detection problem setup in Sec. 6.1.

---

> > ### Comment · Reviewer_QcNM · 2025-05-27
> >
> > My concerns have mostly been addressed. I thank the authors for their thoughtful rebuttal!

---

### Review · Reviewer_CqRc · 2025-04-27

**Summary Of Contributions:**

The paper proposes a new lightweight method for out-of-distribution (OOD) detection based on a novel upper bound for the Overlap Index (OI) between distributions. The key idea is to derive a computationally efficient confidence score using two terms: the distance between the means of the in-distribution (ID) samples and the test sample. The method is theoretically motivated, non-reliant on explicit density estimation, and scalable to high-dimensional feature spaces.

**Audience:**

Yes

**Broader Impact Concerns:**

This is already discusses in the paper under section 6.

**Claims And Evidence:**

No

**Requested Changes:**

1. Clarify the "non-parametric" claim:
Modify the abstract and introduction to note that k (number of bins) is a tunable parameter and affects performance.

2. Adjust the contaminated dataset step:
Instead of building the contaminated mean vector from mixed ID and OOD samples, use only the ID samples to compute the mean vector. Then re-run the "Extra Information" experiments and compare fairly against baselines.

3. Hyperparameter tuning:
Report the grid search details or ranges for all hyperparameters used when fine-tuning baselines.

4. Appendix:
In the appendix, the section headings are formatted as ".1", ".2", etc. It would improve clarity and consistency if these were labeled as "A.1", "A.2", etc., which is more conventional

**Strengths And Weaknesses:**

### Strengths

  1. Theoretical grounding:
    The OI-based confidence score has clear ties to probability theory (Overlap Index and Total Variation Distance). Mathematical properties like insensitivity to small shifts and robustness against Huber-ϵ contamination are proven.

  2. Simplicity and low computational overhead:
    The method does not require training a deep model, inverting matrices, or complex optimization procedures.

   3. Good empirical results:
    The proposed detector shows competitive to superior performance compared to strong baselines across different benchmarks (UCI datasets, CIFAR-10/100, ImageNet-1K).

### Weaknesses

1.  Use of contaminated datasets in some evaluations:
When evaluating under "Extra Information" settings (e.g., Table 2), the method uses the mean of a contaminated dataset containing both ID and OOD samples. This setup advantages the method unfairly and deviates from the standard OOD detection setting. Other techniques like Outlier Exposure [1] (which assume access to auxiliary OOD samples) would be more appropriate comparisons in this case. But my recommendation would be to only use the ID mean rather than a contaminated mean, this would be a more fair comparison.

2.  Missing statistics:
    The results reported are not average over multiple seeds/runs thus mean and std are missing for some of the baselines. The proposed method lacks variability but the baselines are known to exhibit performance fluctuations across runs. Reporting the full performance bands (mean ± std) for all methods would provide a more robust comparison.

3.  Non-parametric claim is misleading:
    While the abstract advertises the method as "non-parametric," the choice of the number of bins $k$ and the indicator functions $g_j$ introduce explicit hyperparameters. Therefore, the detector has parametric tuning hidden in the choice of $g$ and $k$.

4.  Shift vulnerability and weak separation when ID and OOD clusters are moved toward the origin:
    The method heavily relies on the mean vector and norms of samples. If ID and OOD samples are centered near the origin, the detector’s ability to separate them collapses, as acknowledged by the authors themselves.

---
[1] Hendrycks et al. Deep Anomaly Detection with Outlier Exposure ICLR 2019

---

> ### Author Response · Authors · 2025-05-11
> **Weakness 1**
>
> **Response**
>
> As per your suggestion, we have compared our approach with [\*\*], which utilizes outlier exposure for the same task as ours and is authored by the same first author of [1]. The average performance of [\*\*] is 95.6%, while ours stands at 96.2%. Therefore, our approach is comparable to the outlier exposure method.
>
> Moreover, we would like to argue that the assumption of accessing a contaminated dataset containing both ID and OOD samples is valid and fair. In fact, the work of [\*] also uses an unlabeled dataset, where the samples come from the mix of ID and OOD distributions, and compares it with other approaches. Furthermore, compared to [\*], which requires actual OOD samples from the dataset, our approach only requires the mean of the contaminated dataset, making it less restrictive and more practical. We have addressed this in our updated manuscript. However, per your request, we have rerun our approach using only the in-distribution mean. The AUROC for each class, with each class used as the ID class, is as follows:
>
> Plane: 86.24%
>
> Car: 92.49%
>
> Bird: 69.56%
>
> Cat: 79.17%
>
> Deer: 89.26%
>
> Dog: 75.41%
>
> Frog: 87.24%
>
> Horse: 86.97%
>
> Ship: 89.76%
>
> Truck: 90.41%
>
> Ave.: 84.65%
>
> Our approach still shows strong performance, outperforming LPIPS and Bergman, whose average AUROC performances are 79.1% and 82.0%, respectively.
>
> [\*] Choi, Caroline, et al. "Conservative Prediction via Data-Driven Confidence Minimization." Transactions on Machine Learning Research, 2024.
>
> [\*\*] Hendrycks, Dan, et al. "Using self-supervised learning can improve model robustness and uncertainty." Advances in neural information processing systems 32 (2019).
>
> **Action**:
>
> We have mentioned in Sec.4.2 that similar assumption is also used in [\*]. We also added in the “extra information setting” paragraph that our approach is comparable to outlier exposure.

---

> ### Author Response · Authors · 2025-05-11
> **Weaknesses 2, 3, 4, & Request Changes**
>
> **W2**:
>
> We have added means and standard deviations (with 10 runs) to the baseline performance to account for fluctuations in the results in Tables 2 & 5.
>
> **W3**:
>
> ***Response***:
>
> In this context, we use the term non-parametric in the sense that it makes no assumptions about the underlying data distribution. According to Wikipedia [a] and other research works [**], non-parametric methods do not assume a specific form for the data distribution. Our approach aligns with this definition, making it a non-parametric method.
>
> [a] https://en.wikipedia.org/wiki/Nonparametric_statistics
>
> [**] Sun, Yiyou, et al. "Out-of-distribution detection with deep nearest neighbors." International Conference on Machine Learning. PMLR, 2022.
>
> We do not mean that the method does not have hyper-parameters that can be used for tuning.
>
> ***Action***:
>
> We have clarified the terminology about the method being non-parametric in the introduction and mentioned that we are not saying that the method does not have hyper-parameters that can be used for tuning in footnote 1.
>
> **W4**:
>
> We acknowledge this limitation, and improving it will be part of our future work. However, we believe that the strengths of our approach outweigh this limitation.
>
> **Request Changes**:
>
> 1. See our response to **W3**.
>
> 2. See our response to **Weakness 1**.
>
> 3. When using the baselines, we directly downloaded the code from GitHub and followed the provided instructions to run their approach. For methods that require training a model, we utilized the pretrained models made available by the authors. This approach allowed us to achieve their best results. We have added further clarifications in Sec. 4.1.
>
> 4. Done

---

### Decision · Action_Editor_C23Q · 2025-06-19

**Recommendation:** Reject

**Audience:**

Yes

**Audience Explanation:**

Researchers and practitioners working on out-of-distribution detection and non-parametric models might be interested in reading this paper.

**Claims And Evidence:**

No

**Claims Explanation:**

**Summary:**

This paper proposes a lightweight method for out-of-distribution (OOD) detection based on a theoretically motivated confidence score derived from an upper bound of the Overlap Index (OI), defined as one minus the total variation distance between two distributions. The key idea is to compute a non-parametric confidence score using the distance between the test sample and the mean of in-distribution (ID) samples, offering a scalable and interpretable alternative to existing approaches. Unlike deep OOD detectors, which often suffer from high computational costs, the need for hyperparameter tuning, and limited interpretability, the proposed OI-based method is computationally efficient and easy to implement. Empirical evaluations across representative datasets show that the proposed method performs on par with or better than several OOD detectors.


**Claims:**

The key claims made in the paper are that the proposed OI-based OOD detector (1) is novel, non-parametric, lightweight, and easy to interpret, hence providing strong flexibility and generality; (2) competitive with state-of-the-art OOD detectors in terms of detection accuracy on a wide range of datasets while requiring less computation and memory costs; and (3) inherits theoretical robustness properties from OI.


**Evidence:**

The evidence presented in the paper generally supports the proposed claims, but the support for Claim 2 – regarding empirical performance – is relatively weak. As Reviewer KgHs pointed out, “the experimental results are not really impressive. In Table 5, the proposed method is only the best on SVHN. Although the method has the highest average performance, this is largely because its performance on SVHN is much higher than other compared methods (though I cannot see why this should be the case). If SVHN is neglected then the proposed method is not the best.”

Additionally, Reviewer KgHs pointed out that “a large number of OOD detection methods have been proposed in recent years. The newest methods compared in this paper are from 2022, which I am not sure is the state of the art,” raising concerns about the lack of comparison with more recent baselines. While the revision includes a comparison with a newer method (DCM), which outperforms the proposed approach, the authors argue that their method offers advantages in training time, memory usage, and data requirements. These efficiency claims would be more convincing if supported by quantitative comparisons.

Furthermore, given the inconsistent performance across datasets, especially the disproportionately high results on SVHN, a deeper analysis is needed to understand when and why the proposed method excels. Such insight is essential for guiding practical deployment.

Although the revised version adds experiments under image corruption settings, results for relevant baseline methods are missing, limiting interpretability of the findings.

Regarding Claim 3 on theoretical contributions, reviewers noted that the analysis is relatively modest. In particular, Reviewer KgHs remarked that the presented bound is a straightforward derivation from the definition, rather than a substantial new theoretical insight.

**Meta-Review:**

After the authors’ revision, two reviewers recommended “leaning accept” and one recommended “leaning reject.” However, the paper lacks a strong advocate. Remaining concerns primarily center on the weak empirical evidence and the modest theoretical contributions.

Reviewer KgHs emphasized that the experimental results are not compelling, with gains largely concentrated on specific datasets (e.g., SVHN in Table 5), raising questions about generalizability. The analysis lacks depth, and comparisons with recent baselines are insufficient to thoroughly validate the method’s effectiveness. Although additional settings, such as image corruption, were explored, the evaluation remains incomplete.

Moreover, the theoretical contributions are not well distinguished or rigorously validated, and the writing requires improvement for clarity.

Based on these points, I recommend rejection to encourage the authors to further improve their method. They need to convincingly demonstrate the significance of their work and enhance the comprehensiveness of the evaluation.

**Resubmission Of Major Revision:**

The authors may consider submitting a major revision at a later time.